# Tailoring the Techno-Functional Properties of Fava Bean Protein Isolates: A Comparative Evaluation of Ultrasonication and Pulsed Electric Field Treatments

**DOI:** 10.3390/foods13030376

**Published:** 2024-01-24

**Authors:** Saqib Gulzar, Olga Martín-Belloso, Robert Soliva-Fortuny

**Affiliations:** 1Department of Food Technology, Engineering and Science, University of Lleida, Avda. Rovira Roure 191, 25198 Lleida, Spain; olga.martin@udl.cat (O.M.-B.); robert.soliva@udl.cat (R.S.-F.); 2Agrotecnio CERCA Center, Avda. Rovira Roure 191, 25198 Lleida, Spain

**Keywords:** ultrasonication, pulsed electric fields, fava bean protein isolate, protein functionality, plant-based proteins

## Abstract

The fava bean protein isolate (FBPI) holds promise as a sustainable plant-based protein ingredient. However, native FBPIs exhibit limited functionality, including unsuitable emulsifying activities and a low solubility at a neutral pH, restricting their applications. This study is focused on the effect of ultrasonication (US) and pulsed electric fields (PEF) on modulating the techno-functional properties of FBPIs. Native FBPIs were treated with US at amplitudes of 60–90% for 30 min in 0.5 s on-and-off cycles and with PEF at an electric field intensity of 1.5 kV/cm with 1000–4000 pulses of 20 μs pulse widths. US caused a reduction in the size and charge of the FBPIs more prominently than the PEF. Protein characterization by means of SDS-PAGE illustrated that US and PEF caused severe-to-moderate changes in the molecular weight of the FBPIs. In addition, a spectroscopic analysis using Fourier-transform infrared (FTIR) and circular dichroism (CD) revealed that US and the PEF induced conformational changes through partial unfolding and secondary structure remodeling from an α-helix to a β-sheet. Crystallographic and calorimetric determinations indicated decreased crystallinity and lowered thermal transition temperatures of the US- and PEF-modified FBPIs. Overall, non-thermal processing provided an effective strategy for upgrading FBPIs’ functionality, with implications for developing competitive plant-based protein alternatives.

## 1. Introduction

*Vicia faba* L., commonly known as fava bean, broad bean, or horse bean, is an economically important legume crop belonging to the Fabaceae family. Based on current estimates, the global fava bean market value reached USD 3.18 billion in 2021 and is projected to increase to USD 3.47 billion by 2025 [1]. This market expansion largely reflects the growing worldwide demand for nutrient-rich natural and plant-based proteins. Fava beans contain approximately 20–35% protein, with a superior amino acid profile than that of common cereal grains due to a higher lysine content [2]. As health, environmental, and ethical concerns associated with meat consumption rise, plant-based proteins are gaining popularity as alternatives. In 2021, the global plant-based protein market was valued at USD 13.1 billion, and it is projected to surpass the value of USD 22.7 billion by 2031 at a CAGR of 5.7% [3]. Recent advancements in extraction techniques, processing methods, and genetic engineering have expedited the commercial development and incorporation of this legume into meat analogue or meat substitute products [4]. Overall, fava beans show promise as a sustainable, functional plant protein ingredient capable of satisfying escalating demand.

Generally, native plant proteins often exhibit limited gelling and emulsifying capabilities in food systems due to their compact globular tertiary structures, which restrict functional properties, particularly at a neutral pH [5]. The functionality of plant proteins is limited compared to animal proteins, restricting their use in foods. Fava bean proteins demonstrate adequate physicochemical and techno-functional properties for food applications but exhibit lower solubility and gelation than soy proteins [6]. These functions, however, are dependent on the type of protein, its chemical composition, amino acid sequence, and secondary and higher-order structures [2]. A protein’s three-dimensional configuration dictates its functional properties, which can be modified by subjecting the protein to physical, chemical, or biological treatments which change its structure and corresponding functions. Partial denaturation and controlled aggregation enhance the solubility, heat stability, and foaming and emulsifying abilities of plant-based proteins [7,8,9]. It is pertinent to understand the process behind the structure–function relationships of dietary proteins and how they can improve the quality and functionality of proteins or protein-rich food products. Several technical approaches, including high-pressure homogenization [10], enzymatic hydrolysis [11], and transglutaminase treatment [12], have been investigated so far to increase the solubility and functional modification of fava bean proteins. Some treatments, such as transglutaminase, have failed to improve fava bean proteins’ functional attributes and solubility, while high-pressure homogenization and enzymatic hydrolysis, despite improving their solubility, had a negative effect on their emulsifying properties [13]. Modification techniques are vital in enhancing functionality without compromising nutrition. Non-thermal techniques offer advantages over conventional heating and chemical treatments by preserving sensory attributes, essential amino acids, and digestibility [14], thus providing a promising route for upgrading plant proteins.

Ultrasound (US) and pulsed electric fields (PEF) have gained increasing interest as non-thermal techniques [15,16]. Several studies demonstrate that these techniques increase surface hydrophobicity and the exposure of sulfhydryl groups in soy and whey proteins [17,18,19]. Ultrasound induces cavitation, shear, and radical formation, modifying proteins via unfolding, aggregation, and bond breakdown processes [20]. It has been reported that ultrasound-assisted alkaline shifting efficiently improves the solubility of FBPIs [21]. PEF applications of short high-voltage pulses cause electric field-induced protein reorientation, altering protein conformation [22]. Overall, US and PEF show potential for augmenting plant protein functionality through induced structural changes. While prior studies have evaluated the ultrasonication-assisted treatment of FBPIs at an alkaline pH to modify the proteins’ functionality, they have been limited by challenges in precisely controlling the cavitation intensity and achieving uniform effects throughout the sample volume. In contrast, PEF processing allows for a more homogeneous disruption across the particulate network through the direct and targeted application of electric fields without added chemical reagents. This study, therefore, not only presents a novel non-thermal protein modification method for FBPIs, but also overcomes the limitations of heterogeneous impacts associated with ultrasonication alone. Given PEF’s ability to induce non-thermal effects through intense pulsed voltages, a comparative analysis with ultrasound could provide insights into its efficacy for functional modification. Therefore, the current study aimed to enhance key techno-functional properties of FBPIs, including solubility, emulsifying activity, emulsion stability, and foaming capacity, through PEF and ultrasonication treatments, especially at a neutral pH. By characterizing their effects through standardized assays, this research sought to elucidate the structure–function relationships governing plant protein behaviors and assess each technique’s potential for developing FBPIs into a competitive plant-based protein ingredient with diverse food/supplement applications. Our findings may further elucidate the benefits of non-thermal processing for “upcycling” underutilized fava beans through value-added product innovation.

## 2. Materials and Methods

### 2.1. Materials

Fava bean flour was procured from Harinas El Molino (Granada, Spain). The flour was sieved through an 80 mesh sieve (0.18 mm diameter) to remove the coarse particulate matter. The collected flour was stored at −40 °C. Hexane (98% purity) and isopropanol (99.9% pure) were procured from LabKem (Barcelona, Spain). 1-anilinonaphthalene-8- sulphonic acid (98% purity) and 5,5′-dithiobis (2-nitrobenzoic acid) (DTNB) (>98% pure) were purchased from Sigma–Aldrich (St. Louis, MO, USA). The SDS-PAGE chemicals were procured from Bio-rad (Hercules, CA, USA). All the reagents used in this study were of analytical grade.

### 2.2. Preparation of Fava Bean Protein Isolate

The fava bean protein isolate (FBPI) was prepared based on the method described by Gulzar et al. [9] Briefly, fava bean (*Vicia faba* L.) flour was subjected to solvent defatting using a mixture of hexane and isopropanol at a flour-to-solvent ratio of 1:5 (*w*/*v*) with constant stirring for 2 h at room temperature. After solvent evaporation, the defatted flour was dispersed in deionized water (DW) at 5% (*w*/*v*) and adjusted to a pH of 9.5 ± 0.1 using 1 M NaOH while stirring continuously for 1 h. The slurry was centrifuged at 10,000× *g* for 15 min at 4 °C (Beckman Coulter, Avanti J-26 XP centrifuge, Brea, CA, USA). The pH of the supernatant was then lowered to 4.5 ± 0.1 using 2 M HCl and re-centrifuged under identical conditions. The precipitate was thoroughly washed with DW until a neutral pH was achieved. Finally, the isolated protein was freeze-dried using a Telstar LyoQuest laboratory freeze-drier (Azbil Corporation, Tokyo, Japan) at −70 °C for 96 h and stored at −40 °C until further analysis.

### 2.3. Ultrasonication (US) Treatment of FBPI

The FBPIs were rehydrated in DW to obtain 10% (*w*/*v*) protein dispersions, and the resultant dispersions were subjected to US treatment using a probe type UP400S (400 W, 24 kHz) ultrasonic processor (Hielscher Ultrasonics GmbH, Teltow, Germany) for 30 min in pulsed mode with duty cycles of 0.5 s on and 0.5 s off at varying amplitudes of 60%, 70%, 80%, and 90%. The energy densities corresponding to the US amplitudes of 60%, 70%, 80%, and 90% were calculated to be 138.6 kJL^−1^, 167.4 kJL^−1^, 194.4 kJL^−1^, and 217.8 kJL^−1^, respectively. The US parameters were selected based on extensive studies of the literature followed by initial trials. The temperature during the US treatment was regulated at 25 °C using a temperature control unit. The treated FBPIs were immediately lyophilized, and the resulting powders were designated as US-FBPI-60, US-FBPI-70, US-FBPI-80, and US-FBPI-90, corresponding to their US treatment at 60%, 70%, 80%, and 90% amplitudes, respectively, and stored at −40 °C.

### 2.4. PEF Treatment of FBPI

The FBPI dispersions (10% *w*/*v*) in DW were subjected to PEF treatment in batch mode using a laboratory-scale PEF system (EPULSUS^®^ LBM1A-15, EnergyPulse Systems, Lisbon, Portugal). The treatment chamber had a working volume of 200 mL with a 1 cm gap between two rectangular stainless steel electrodes (15 cm × 10 cm). The FBPIs were exposed to bipolar rectangular pulses with an electric field intensity of 1.5 kV/cm, a 20 μs width, at a frequency of 20 Hz. The FBPI dispersions were treated with 1000, 2000, 3000, and 4000 pulses. These PEF conditions were selected after extensive initial trials. During processing, the electric current and power output were recorded to be 11.3 A and 42 W per pulse, respectively. The energy densities corresponding to 1000, 2000, 3000, and 4000 pulses were calculated to be 4.2 kJL^−1^, 16.8 kJL^−1^, 37.8 kJL^−1^, and 67.2 kJL^−1^, respectively. After each treatment, the treated FBPI was immediately freeze-dried and stored at −40 °C. The obtained powders were designated as PEF-FBPI-1000, PEF-FBPI-2000, PEF-FBPI-3000, and PEF-FBPI-4000, corresponding to their PEF treatment at different pulse amounts, respectively. The control FBPI without any treatment was designated as CON-FBPI.

### 2.5. Determination of Functional Properties

#### 2.5.1. Protein Solubility

Solubility testing of the US- and PEF-treated FBPIs was carried out using the modified Biuret method as adopted by Gulzar et al. [9]. Briefly, FBPI samples (10 mg/mL) were dispersed in DW and stirred using a magnetic stirrer for 1 h at 1000 rpm and centrifuged at 10,000× *g* for 20 min at room temperature using a Hettich EBA 21 centrifuge (Hettich GmbH, Tuttlingen, Germany). One part of supernatant was mixed with five parts of Biuret reagent and incubated for 30 min at room temperature. The absorbance was read at 540 nm against a reagent blank using the Jenway 6850 UV/Vis spectrophotometer (Thermo Fisher Scientific, Hampton, VA, USA). A standard curve of concentration versus absorbance was constructed using bovine serum albumin (BSA) as a standard to estimate the protein concentration of the FBPIs. Protein solubility was expressed as the percentage of protein remaining in the solution.

#### 2.5.2. Surface Hydrophobicity and Reactive Sulfhydryl (SHr) Content

The surface hydrophobicity of the FBPI was determined using a 1-anilinonaphthalene-8-sulphonic acid (ANS) fluorescent probe according to the method described by Benjakul and Morrissey [23]. Briefly, the FBPI was dissolved in a phosphate buffer (pH 7.0) and incubated with 60 μM of ANS solution for 15 min. The fluorescence intensity was measured at excitation and emission wavelengths of 390 nm and 470 nm, respectively, using a fluorescence spectrophotometer (Tecan Infinite M200, Tecan, Grödig, Austria). Surface hydrophobicity was expressed as the initial slope of the fluorescence intensity against the protein concentration.

The SHr content of the FBPI was measured using 5,5′-dithiobis(2-nitrobenzoic acid) (DTNB) as per the method of Beveridge et al. [24]. The FBPIs were solubilized in 50 mM of phosphate buffer (pH 7.0) and reacted with Ellman’s reagent (4 mg/mL DTNB in Tris-glycine buffer). Absorbance was read at 412 nm using a 0.6 M KCl solution as a blank, and the SHr content was calculated using the molar extinction coefficient of 13,600 M^−1^ cm^−1^.

#### 2.5.3. Emulsifying Properties

The emulsifying activity index (EAI) and the emulsion stability index (ESI) of the FBPI suspensions in DW at varying concentrations (0.1%, 0.5%, 1%, and 3% *w*/*v*) were evaluated using a modified method based on Pearce and Kinsella [25]. The FBPI suspensions (15 mL) were homogenized with 10 mL of soybean oil at 20,000 rpm for 1 min. Aliquots (50 μL) were drawn from the bottom of the emulsions at 0 and 30 min and diluted 100-fold using 0.1% SDS solution. Finally, the absorbance of the diluted samples was measured at 500 nm and used for the EAI and ESI calculations (Equations (1) and (2)):(1)$$EAI\;(m^{2}/g)=\frac{2\times 2.303\times A_{0}\times DF}{C\times L\times (1-\emptyset )\times 10,000}$$
(2)$$ESI\left(min\right)=\frac{A_{0}\times ∆t}{A_{0}{-A}_{30}}$$
where A_0_ and A_30_ are the absorbance taken at 0 and 30 min, respectively. DF is the dilution factor; C is the initial protein concentration (g/mL); L is the cuvette path length (m); ∅ is the oil volume fraction; and Δt = 30 min.

#### 2.5.4. Foaming Properties

The foaming properties of the FBPIs subjected to different treatments were determined through their foaming ability (FA) and foaming stability (FS) following the modified method of Shahidi et al. [26]. The FBPIs were prepared by rehydrating the powders in DW to obtain a concentration of 10 mg/mL. An aliquot of 20 mL from each sample was precisely transferred to a 100 mL graduated cylinder. The solutions were homogenized at 13,400 rpm for 1 min at room temperature using a high-speed homogenizer (IKA T-25 Ultra Turrax, IKA-Werke GmbH, Staufen, Germany) to induce foaming. The FA and FS were calculated using Equations (3) and (4):(3)$$FA\left(\%\right)=\frac{V_{T}}{V_{0}}\times 100$$
(4)$$FS\left(\%\right)=\frac{V_{t}}{V_{0}}\times 100$$
where V_T_ is the total volume after homogenization; V_0_ is the original volume before homogenization; and V_t_ is the total volume after leaving the sample at room temperature for 60 min.

### 2.6. Characterization of FBPI Powder

#### 2.6.1. Particle Size and Zeta Potential

The particle size distribution of the FBPI was determined using static light scattering (Mastersizer 2000, Malvern Panalytical Ltd., Malvern, UK). The FBPI was diluted in 10 mM of phosphate buffer (pH 7.0) before the analysis. The light scattering intensity data were collected and processed using the Mastersizer software (version 3.62) to derive average particle diameters reported as the volume-weighted mean diameter (d32). The zeta potential (ζ-potential) was measured using the phase analysis light scattering (PALS) technique on a Zetasizer NanoZS (Malvern Instruments Ltd., Malvern, UK). The FBPIs were diluted 10-fold with a buffer and loaded into a folded capillary cell equipped with gold-plated electrodes. Electrophoretic mobility measurements were made and converted to ζ-potential values using the Smoluchowski approximation model incorporated into the Zetasizer software (version 3.30).

#### 2.6.2. Sodium Dodecyl Sulfate-Polyacrylamide Gel Electrophoresis (SDS-PAGE)

SDS-PAGE was performed based on the method described by Laemmli [27] to determine the protein patterns of the FBPIs. The protein suspensions were dissolved in 5% *w/v* SDS solution and heated at 95 °C for 1 h followed by centrifugation at 7000× *g* for 10 min at 25 °C. The supernatants were mixed with a Laemmli sample buffer containing 2% SDS, 10% glycerol, and 0.05% bromophenol blue in 0.5 M Tris–HCl (pH 6.8) under non-reducing and reducing (+5% β-mercaptoethanol) conditions. Finally, the proteins (15 µg) were loaded onto the polyacrylamide gel (4% stacking gel; 12% running gel). Electrophoresis was performed at a constant current of 15 mA/gel until completion, followed by staining with Coomassie Brilliant Blue R-250 and destaining. Precision Plus ProteinTM dual color standards (Bio-Rad) were run concurrently for molecular weight estimations.

#### 2.6.3. Fourier-Transform Infrared (FTIR) Spectroscopy

The attenuated total reflectance (ATR) FTIR spectroscopy of the FBPI was conducted on a Jasco FT/IR 6300 spectrometer (Jasco Inc., Tokyo, Japan) equipped with a PIKE MIRacle ™ ATR sampling accessory containing a Diamond/ZnSe crystal. Powdered FBPIs were placed on the ATR crystal, and the infrared spectra from the 4000–600 cm^−1^ wavenumber region were collected in 64 scans at a resolution of 4 cm^−1^. The raw spectra were processed and analyzed using the SpectraManager software version 2.8 (Jasco Analitica Spain S.L, Madrid, Spain).

#### 2.6.4. Circular Dichroism (CD) Spectroscopy

The CD spectra of the FBPIs were recorded on a Jasco J-810 spectropolarimeter (Jasco Inc., Tokyo, Japan) equipped with a thermostatically controlled cell holder. FBPI solutions were prepared at 0.03 mg/mL in 0.01 M of phosphate buffer (pH 7.2) such that absorbance was below 1.0. The spectra from 200 to 260 nm were collected at a 100 nm/min scan speed, with a 2 s response time, and a 1 nm bandwidth under a constant nitrogen purge. Protein secondary structure estimation was performed using the SpectraManager software (Version 2, JASCO, Tokyo, Japan).

### 2.7. Thermal Properties

A differential scanning calorimeter (Mettler Toledo STARe SYSTEM DSC 3+, Columbus, OH, USA) was used to conduct the thermal analysis of the FBPIs. The FBPIs were accurately weighed and loaded onto an aluminum crucible of a 40 μL volume, which was then sealed and placed in the DSC chamber along with an empty reference crucible under a continuous nitrogen purge. Temperature scanning was performed at 10 °C min^−1^ over the range of 0–250 °C. The onset melting/solidification temperatures and the associated latent heat values were obtained from the thermograms using the STARe software (version 14).

### 2.8. X-ray Diffraction (X-RD)

An X-RD analysis was conducted at room temperature using a RIGAKU diffractometer (Model RU300, Rigaku Corporation, Tokyo, Japan) equipped with a rotating copper anode X-ray tube operated at 40 kV and 80 mA. The diffractometer utilized Cu Kα radiation (λ = 1.5418 Å) and a graphite monochromator to obtain high-intensity diffraction patterns in the 2θ range of 3–70° with a step size of 0.03° and a dwell time of 1 s per step. Phase identification was performed using the JADE (Materials Data Inc., Livermore, CA, USA) qualitative X-ray analysis software.

### 2.9. Statistical Analysis

A completely randomized design was employed for this study, with all experiments conducted in triplicate (n = 3). The independent batches and the results were reported as the mean ± standard deviation (SD). The data collected were subject to a statistical analysis using SPSS (version 25.0, IBM, New York, NY, USA). A one-way analysis of variance (ANOVA) was performed to determine the significance of effects between the treatment groups. Duncan’s multiple range test was used for mean comparison, and the *t*-test was used for paired comparison.

## 3. Results

### 3.1. Effect of US and PEF Treatment on the Functional Properties of FBPI

#### 3.1.1. Solubility

The solubility of proteins plays a key role in various applications by influencing their emulsification, foaming, and gelation properties [28]. As shown in Table 1, the FBPIs’ solubility was enhanced through the US and PEF treatments in a magnitude-dependent manner. The solubility of the CON-FBPI was 74.38 ± 2.16%, while, for the US-treated FBPI, the solubility increased to 90.98 ± 2.67%, which was achieved at an ultrasonication amplitude of 70%, while, for the PEF-treated FBPI, the highest solubility of 82.12 ± 1.99% was attained with 2000 pulses, beyond which decreases occurred. The US treatment was more effective in improving the solubility of FBPI compared to the PEF treatment. Partial protein unfolding likely underpinned the initial solubility gains via hydrophilic domain exposure and hydration enhancement. The US treatment is known to cause shear forces and localized hotspots, disrupting protein structures [29]. Higher magnitudes induced denaturation and aggregation, reducing solubility. At moderate amplitudes, such disruption may impart a modest degree of unfolding, favorably exposing a greater number of polar amino acid residues and enhancing solubility through improved hydration mediated by protein–water interactions [30]. However, higher amplitudes can over-unfold proteins, stripping away their native structures. This denatures secondary/tertiary conformations and exposes hydrophobic amino acid residues. Such hydrophobic moieties aggregate into insoluble complexes with restricted hydration, diminishing protein solubility [31]. Sharma et al. [32] reported that the solubility of rice bran protein isolates decreased at a higher ultrasonic amplitude. In a similar study by Rahman et al. [33], it was reported that the ultrasonication of soy proteins at a higher power reduced the solubility of the proteins by forming insoluble aggregates. In the case of PEF, similar trends have been reported for soy protein and mung bean protein treated with PEF, where intensities above the thresholds generated aggregates instead of solubility increases [9,34]. Though PEF’s precise mechanism remains unclear, accepted hypotheses involve peptide dipole moment polarization and unfolding influencing hydration/solubility [35]. Most plant proteins exhibit a poor neutral pH solubility, limiting food applications [36]. Based on our findings, the application of US and PEF treatments could be the ideal strategy for enhancing the solubility of plant-based proteins under near-neutral pH conditions.

#### 3.1.2. Surface Hydrophobicity and SHr Content

The surface hydrophobicity (H_0_) of proteins is considered an essential factor in assessing the number of hydrophobic groups on their surface in the vicinity of a hydrophilic medium. This parameter plays a key role in the conformational structure and functionality (emulsification, foaming, and gelation) of proteins [37]. As shown in Table 1, the H_0_ of the FBPIs subjected to US treatment exhibited a trend of increasing magnitude-dependently with rising amplitudes (*p* < 0.05). Compared to the CON-FBPI, the H_0_ of US-FBPI-70 was increased by 40.53%. However, a substantial decline in H_0_ was observed for the FBPIs treated at amplitudes above 80%. The general ascent in H_0_ values with escalating ultrasonic intensity up to an 80% amplitude suggests that the US successfully exposed additional buried hydrophobic domains within the protein structure. However, the excessively energetic cavitation events induced at a 90% amplitude may have exceeded the thresholds conducive to easy unfolding, potentially favoring reaggregation or the precipitation of hydrophobic moieties. This could explain the subsequent slight decrease in H_0_, representing a shift in the unfolding aggregation balance triggered by an overly forceful acoustic cavitation. Ultrasonication due to the generation of strong cavitational effects can effectively expose the buried hydrophobic regions situated in the interior of the proteins to the hydrophilic surrounding medium [38]. Previous research has shown that US can augment the H_0_ of soy protein isolates [39], soybean glycinin [40], pea protein isolates [41], and black bean protein isolates [42] by exposing buried hydrophobic domains. While US typically boosts H_0_ with rising intensity/duration, some studies observed reductions under excessive parameters possibly due to heat-induced aggregation forming a physical barrier against further unfolding/diffusion of hydrophobic moieties [43,44].

On the other hand, the PEF-treated FBPIs also showed moderately elevated H_0_ values, especially with rising pulse numbers up to 2000 pulses (*p* < 0.05), beyond which marginal declines occurred. A sizeable increase in the H_0_ of PEF-FBPI-2000 by 28.74% was recorded compared to the CON-FBPI. PEF disrupts native protein structures through various mechanisms like polarization and unfolding, pushing the delicate balance towards the unfolding and exposure of buried hydrophobic groups, thus boosting H_0_ values [34]. Moreover, intense electric field pulses can disrupt weaker non-covalent interactions like hydrogen bonding, hydrophobic interactions, and van der Waals, leading to the structural unfolding and exposure of internal hydrophobic amino acid residues [45]. A reduced H_0_ at 4000 pulses coincided with a diminished solubility, implying protein aggregation driven by a heightened hydrophobicity. Similar observations have been reported for soy protein subjected to increasing PEF intensities, where the H_0_ rose initially before diminishing at stronger intensities, a phenomenon attributed to aggregation outweighing unfolding effects [34,46]. Overall, the controlled modulation of proteins’ hydrophobicity–hydrophilicity balance through non-thermal processing represents a mechanistic strategy for optimizing functionality.

The reactive sulfhydryl content (SHr) of the FBPIs is presented in Table 1. The results indicated a similar stepwise rise in the SHr of the US- and PEF-treated FBPIs, aligned with the H_0_ trends. CON-FBPI had a SHr of 10.36 ± 0.34 µmol/g protein, which increased to 15.91 ± 0.27 µmol/g protein at an 80% amplitude for the US-treated FBPI, whereas, for PEF, the highest SHr of 13.56 ± 0.28 µmol/g protein was observed at 2000 pulses. US has been shown to increase SHr groups in various plant proteins including soy protein isolates [17], cross-linked soy proteins [47], and pea protein isolates [41,48], particularly with longer treatment times. Key mechanisms contributing to this effect include cavitation assisting the formation of -SH groups by reducing disulfide bonds, decreasing particle size through the breakdown of intermolecular disulfide linkages, and exposing buried -SH groups on protein surfaces via partial unfolding induced by strong ultrasonic shear forces [38,49]. SHr elevations post-PEF have been reported for other proteins including egg whites, soybeans, and mung beans [9,34,50,51]. Apart from the disruption of non-covalent bonds to expose buried -SH groups, PEF is also hypothesized to modify charge densities around the -COOH and -NH^3+^ moieties of amino acids [52]. The SHr reduction in US-FBPI-90 and PEF-FBPI-4000 possibly reflects a disulfide bond formation between unfolded protein subunits, constraining the free sulfhydryl content.

#### 3.1.3. Emulsifying Properties

The emulsifying activity index (EAI) and the emulsion stability index (ESI) of the FBPIs subjected to varying US and PEF treatments are tabulated in Table 1. Both the US and PEF treatments of FBPI resulted in a considerable upsurge in the EAI and ESI of the protein, by 49.52% in the case of US-FBPI-70 and 36.13% for PEF-FBPI-2000, compared to CON-FBPI. EAI evaluates the ability of proteins to adsorb at oil–water interfaces, quantified as the surface area of the interface covered per unit mass of protein. Specifically, EAI measures the protein’s capacity to localize surrounding oil droplets at the interface. The highest EAI was recorded in the US-FBPI-70, corresponding to its higher solubility and H_0_, which play a pivotal role in the oil-holding capacity of the proteins [53]. The presence of more hydrophobic groups on the surface leads to a considerable decrease in the thermodynamic energy barrier, increasing the flow, collision, rearrangement, and adsorption of protein molecules at the interface [54]. US predominantly modifies particle sizes, while PEF enacts broader intra-molecular changes through electric field interactions, explaining their distinct effects on emulsifying behavior. By unfolding proteins more and potentially aggregating hydrophobic regions, PEF modifies interfacial properties in a manner that promotes both initial emulsification through increased availability of interfacial peptides, as well as long-term stability from reorganized hydrophobic–hydrophilic patterns. This expands its applications in products like baked goods, restructured meats, and ready-to-eat meals. Martínez-Velasco et al. [55] reported that ultrasonication led to a substantial decrease in the surface tension of the FBPI, which improved the adsorption at the oil–water interface. Moreover, the reduction in the particle size of proteins also promotes a better interface activity owing to their better mobility in moving from the aqueous phase into the water/oil interface [21]. The emulsion activity of grass pea protein isolates was significantly improved by ultrasonication, as indicated by enhanced EAI and ESI [30]. Furthermore, PEF treatment has also been proven to improve the emulsifying properties of whey protein isolates [56], canola proteins [57], sunflower protein isolates [7], and mung bean protein isolate [9].

#### 3.1.4. Foaming Properties

The foaming properties expressed as foaming ability (FA) and foaming stability (FS) of the FBPIs as affected by the US and PEF treatments are tabulated in Table 1. The FA of the FBPIs was found to be drastically improved (*p* < 0.05) by both the US and PEF treatments, with the former increasing the FA by 3-fold and the latter by nearly 2-fold. Native FBPIs have relatively poor foaming properties, attributed to their compact globular structure and low flexibility. However, novel approaches like US have been proven to modify FBPIs and enhance their foaming functionality [21,55]. The presence of surface-exposed hydrophobic amino acid residues facilitates interactions between the air and water interfaces, effectively stabilizing foam and emulsion systems. Hydrophobic moieties residing on the protein surface are thought to enhance adsorption kinetics at interfacial boundaries by driving the thermodynamically favorable exclusion from the aqueous phase [58]. Additionally, reduced particle dimensions arising from the ultrasonication treatment result in a higher surface density of the adsorbed protein molecules per unit area. The US- and PEF-aided unfolding of proteins provide more surface-active molecules for faster adsorption to the air–water interfaces and foam formation. Optimum ultrasonication can cause a slight protein degradation, which contributes to smaller peptide fragments which pack tightly at the bubble surface, conferring added stability against foam collapse (3). However, excessive treatment, especially at higher amplitudes, reduces molecular weight and flexibility below the optimal levels required for good foaming, which can lead to lower FS, as indicated by the FA and FS of the US-treated FBPIs above a 70% amplitude. On the contrary, the PEF-treated FBPIs had better FS due to the moderate conformational changes enabled by pulsed fields, which assisted in a rapid reduction in surface tension for improved foam generation and expansion. The better foaming properties of FBPIs have the potential to replace egg whites/albumen with cheaper fava bean protein alternatives. Also, they could provide clean-label appeal as plant-based foaming agents without E-number emulsifiers/stabilizers as well as gluten-free formulations where wheat-based leavening is restricted.

### 3.2. Characterization of FBPI as Affected by Varying US and PEF Treatments

#### 3.2.1. Particle Size and Zeta Potential

Table 2 depicts the particle size parameters of the FBPIs subjected to varying US and PEF treatments. CON-FBPI had an average surface-weighed particle size of 1.20 ± 0.02 µm, represented as the surface mean diameter of the particles. The volume-weighted particle size distributions of the FBPIs are depicted in Figure 1. Ultrasonication led to successive decreases in the average particle size of the FBPIs with the increasing amplitudes. The shear forces and cavitation effects generated during ultrasonication are responsible for the breaking down of large aggregates [48]. Ultrasonication at a 90% amplitude resulted in a 4-fold reduction in the 0particle size of US-FBPI-90, which demonstrates the ability of ultrasound to effectively disaggregate the isolate into smaller fragments based on the disruption of non-covalent linkages. Moreover, CON-FBPI showed a bimodal distribution, with an average volume-weighed particle diameter of 72.6 ± 5.1 µm. The US treatment of the FBPIs resulted in the flattening of the size distribution curve to a multimodal distribution. The PEF-treated FBPIs, however, showed a fairly uniform size distribution with larger volume-weighed particle diameters, plausibly due to aggregation. Alavi et al. [21] reported that the particle size analysis of FBPIs subjected to ultrasonication treatment at a pH of 7 for 10 min revealed a Z-average diameter of 335.3 nm, approximately 6-fold smaller than the untreated FBPI control. Furthermore, when the duration of ultrasonication was extended to 20 min, it resulted in an additional reduction in the mean particle size to 288.5 nm. Mozafarpour et al. [30] also reported a decrease in the particle size of grass pea protein isolates through ultrasonication. On the other hand, the PEF treatment also caused a moderate decrease in the average particle diameter of the FBPIs. The particle size of PEF-FBPI-4000 was reduced by half when the most intense PEF treatment was applied. The pulsing nature of electric field exposure for short timescales causes the breakdown of aggregates into smaller sizes. Overall, submicron diameters represent native storage protein bodies that undergo minor depolymerization by electric pulsing, while ultrasonication drastically reduces particle size owing to its ability to rupture aggregates via strong hydrodynamic shear forces.

The ζ-potential values of the FBPIs ranged from −26.27 mV to −20.10 mV (Table 2), depicting a significant decrease (*p* < 0.05) in the ζ-potential after ultrasonication, particularly at higher amplitudes. Native globular proteins structurally sequester hydrophobic regions including the ionic, aromatic, and alkyl functional groups [55]. The cavitational shear forces of ultrasounds cause a partial unfolding that brings these buried hydrophobic domains outward, thereby altering the net surface charge assessed by means of ζ-potential measurements [59]. Moreover, US treatment has also been shown to result in a significant size reduction in the polymeric chains, which also affects the net charge on the macromolecules [60]. Conversely, the PEF-treated FBPIs showed only a slight decrease in the ζ-potential, with an insignificant change (*p* > 0.05) in the values among the FBPIs treated with varying pulse numbers. The ζ-potential values of the FBPIs were in line with the H_0_ values, in which US-treating FBPIs caused the prominent exposure of buried hydrophobic amino acids, compared to PEF, which resulted in the overall change in the charge of the polypeptide chains.

#### 3.2.2. SDS-PAGE Protein Patterns

The protein patterns of the FBPIs subjected to varying US and PEF treatments, under non-reducing and reducing conditions, are presented in Figure 2a,b, respectively. Under non-reducing conditions, across all FBPIs, three predominant bands were detected at approximate molecular weights of 66 kDa, 55 kDa, and 49 kDa, attributed to the globulin storage protein subclasses of convicilin, legumin, and vicilin, respectively [61]. Additionally, a minor band present at ~74 kDa, possibly corresponding to minor legumin subunits, was detected [62]. Under reducing conditions, the polypeptide band corresponding to legumin (~55 kDa) dissociated into its characteristic component subunits for all FBPIs. Specifically, bands emerged lower in the gel at approximately 35 kDa and 19–23 kDa, attributed to the acidic α-legumin subunit and basic β-legumin subunits, respectively, based on the molecular weights [63]. After US treatment (lanes 2–5), under non-reducing conditions, a reduction in the staining intensity of the vicilin bands compared to CON-FBPI could be seen (especially in US-FBPI-60 and US-FBPI-70), which indicated the partial degradation/fragmentation of vicilin subunits by ultrasonication. However, in US-FBPI-80 and US-FBPI-90, high MW aggregates appeared as smears at the head of the SDS-PAGE gel, and the band intensity across all polypeptides remained unchanged. On the contrary, under reducing conditions, there was a gradual decrease in the band intensity across all FBPIs treated with ultrasonication. The appearance of high MW smears in non-reducing conditions implied that, at higher intensities (80 and 90%), the dissociated polypeptides aggregated via disulfide bonds. Nevertheless, under reduced conditions, these disulfide bonds were broken, revealing that ultrasonication caused more degradation in the polypeptide chains, especially at higher intensities. Alavi et al. [21] reported a similar appearance of high MW smears and loss of band intensity in the SDS-PAGE patterns of FBPIs upon ultrasonic-assisted alkaline treatment, especially at higher intensities. The PEF-treated FBPIs showed a similar but subtler effect on the protein subunit profile compared to the ultrasound treatment. Our findings are in agreement with previous studies by Gulzar et al. and Li et al. [9,34], who also observed no alterations to mung bean protein and soy protein band patterns, respectively, under non-reducing conditions following PEF treatment. However, when analyzed under reducing conditions, a progressive loss of intensity in the polypeptide bands near 66 kDa and 54 kDa with ascending pulse numbers was observed, indicating that disulfide bonds played a profound role in the formation of aggregates. The electrophoretic pattern results were concomitant with the particle size of the FBPIs (Table 2) as affected by the US and PEF treatments, further substantiating that US and PEF caused structural changes in the native polymeric structure of the FBPIs. Overall, it was observed that the US treatment caused more prominent changes in the FBPIs, including the fragmentation of vicilin subunits and the dissociation of legumin, compared to the PEF treatment, where only moderate changes in the MW could be observed.

#### 3.2.3. Fourier-Transform Infrared (FTIR) Spectra

Figure 3 depicts the FTIR spectrum of the FBPIs subjected to varying US and PEF treatments. Characteristic absorption peaks representing the native structural conformation of FBPIs were observed in all the FBPIs. The amide A band between 3410 and3300 cm^−1^ corresponds to N-H stretch vibrations, indicative of hydrogen-bonding interactions [64]. The doublet absorption band centered near 2930 cm^−1^, consistent with the amide-B region, arises from the asymmetric stretching configuration of the O-H and N-H bonds present within the carboxylic acid and ammonium functional groups, respectively [65]. Amide I, appearing from 1700 to 1600 cm^−1^, primarily involves carbonyl (C=O) stretching of the peptide linkage and is sensitive to secondary structures [66]. The amide II band at 1550 cm^−1^ arises from combinations of N-H bending and C-N stretching vibrations. Additional peaks near 1160 cm^−1^ and 990 cm^−1^ correspond to the C-O stretching modes of ester linkages [67]. Upon US treatment, a noticeable decrease in intensity is observed for the amide I and amide II peaks. The reduction in the 1650 cm^−1^ peak suggests that ultrasonication induces a loss of ordered secondary structures due to protein unfolding and the disruption of intramolecular hydrogen bonding. The diminished 1540 cm^−1^ peak indicates changes in the peptides’ backbone structure. Additionally, the N-H stretching peak at 3300 cm^−1^ shows a marked drop in intensity after ultrasonication. This reveals alterations in the hydrogen bonding of N-H groups likely due to conformational changes and aggregation. Martínez-Velasco et al. [55] demonstrated that ultrasonication caused a drastic decrease in the intensity bands of fava bean protein isolates in the amide A, amide I, and amide II regions. Several studies have revealed that the denaturation of proteins resulted in a decrease in band intensities, particularly in the amide I and amide II regions [68,69,70]. The effects are more pronounced with the increasing duration of ultrasonic exposure. In the PEF-treated FBPIs, the reductions in the amide I, amide II, and N-H stretching peaks were much lower compared to those following ultrasonication. The relatively lower loss of native conformation is attributed to the milder effects of PEF in inducing structural changes. Previous studies have also shown that no discernible shifts in peak positions or intensities between proteins suggest that the overall conformation remained unchanged [9]. Therefore, a negligible denaturation resulted from PEF processing under the conditions examined.

#### 3.2.4. Circular Dichroism (CD) Spectra

CD spectroscopy was conducted to investigate the effects of the US and PEF treatments on the secondary structural composition of the FBPIs. Figure 4a shows the CD spectra of the FBPIs treated with varying US and PEF treatments. The CON-FBPI spectra showed a broader positive band at 200 nm, which is indicative of α-helical content, while the negative peak at 218 nm corresponds to β-sheet content. Moreover, it can be observed from the spectra that the US and PEF treatments caused successive declines in the α-helical bands at 200 nm in the order of magnitude of the treatments, which substantiates the fact that the US treatment induced significant alterations to the secondary structure of the FBPIs, followed by the PEF treatment. Further quantification of the secondary structures, as expressed in Figure 4b, demonstrated that the ultrasound treatment, in a magnitude-dependent order, reduced the α-helical structure while conversely increasing the β-sheet content of the FBPIs. The US treatment caused a 10.93% reduction in the α-helical content of the native FBPIs at a 90% intensity, whereas the PEF at 4000 pulses reduced the α-helical content by 4.54%. The loss of α-helical content resulted in the subsequent rise in the β-sheet content of the FBPIs. The disruption of the native α-helical motifs and the partial conversion to a β-conformation is postulated to arise from the ultrasonic cavitation forces unfolding the polypeptide chains. Our findings were in agreement with Mozafarpour et al. [30] and Malik et al. [7] regarding the relationship between ultrasonication and protein secondary structures’ propensities, particularly with increasing intensities. In a study by Stathopulos et al. [71], it was observed that aggregate formation coincided with decreases in the α-helical content and increases in the β-sheet content following ultrasonication. PEF was rather moderate in the structural modification of the FBPIs. Nevertheless, a secondary structure analysis via the CD analysis of soy protein isolates treated with PEF has shown a reduction in the α-helix content and an increase in the β-sheet content [34]. Collectively, these findings illustrate how the US and PEF treatments can differentially induce secondary structural remodeling depending on the balance of the cavitational effects and nonlinear interactions between electric fields and protein conformations.

### 3.3. X-ray Diffraction (XRD)

An XRD analysis was conducted to characterize the changes in the crystalline structure of the FBPIs treated with US and PEF at varying intensities. The XRD spectra, delineated in Figure 5, exhibited two peaks near 2θ = ~10° and 2θ = ~20° for all the FBPIs, which have been labelled crystalline region I (2θ = 10°) and crystalline region II (2θ = 20°), indicative of ordered structural domains within the protein [72]. Nevertheless, the US- and PEF-modified FBPIs recorded changes in the intensity and slight variations in the 2θ in these two crystalline regions. The dip in the intensity of the proteins is indicative of the loss of relative crystallinity [73]. Diffraction intensity decreased progressively with increasing US amplitudes and PEF pulse numbers. Moreover, the reduction in crystal size also correlates inversely with the size of the particles. Therefore, the reduced particle size of the FBPIs aided by US and PEF treatments corroborates the loss of relative crystallinity. Similar observations of an amorphous halo presenting as a peak at 2θ = 20° have been reported for ultrasound-modified soy protein isolates [74]. Malik et al. [7] also documented a decrease in the diffraction intensity of sunflower protein isolates post ultrasonication. The degree of crystallinity relates to functional properties like solubility and water-holding capacity, and it is a useful index in the texture engineering of foods. In the case of proteins, it provides insights into supramolecular assemblies and the glass transition behavior that influences rheology. XRD delivers quantitative structural insights applicable to new product development, formulation optimization, food processing efficiency, and quality management efforts that use proteins.

### 3.4. Thermal Properties

The thermal properties of the FBPIs, characterized using differential scanning calorimetry (DSC), as affected by the US and PEF treatments, are shown as DSC thermograms in Figure 6. Along with the tabulated onset/peak/endset temperatures and enthalpy values (Table 3), the thermograms show that the FBPIs underwent major changes in their denaturation temperatures after the US and PEF treatments. In particular, the US treatment of the FBPIs lowered the denaturation temperature, suggesting weakened van der Waals and hydrogen bonding [75]. The peak thermal transition temperature of CON-FBPI was reduced from 166.63 °C to 150.09 °C for US-FBPI-70. Moreover, broader peaks were identified in the US-treated FBPIs, reflecting unfolded domains, that melted over a wider range of temperatures due to increased flexibility [76], whereas lower enthalpy values signify fewer intact intramolecular bonds after US-induced unfolding [77]. The enthalpy value of CON-FBPI declined from 191.86 J/kg to 112.75 J/kg for US-FBPI-90. On the other hand, the PEF treatment caused minor changes in the thermal properties due to its mild effects on the protein structure, with the highest decline of 4.07 °C in the peak thermal transition temperature observed for PEF-FBPI-4000. The information obtained by characterizing the thermal transition behavior of the proteins can be used for tailored protein functionality and applications.

## 4. Conclusions

Non-thermal processes such as US and PEF can be used to modulate the structure–function relationship and enhance the techno-functional properties of FBPIs, hence offering insights into optimizing plant proteins’ functionality for developing more sustainable and clean-label food products. Relevant properties of FBPIs, namely, its solubility, surface hydrophobicity, and emulsifying and foaming capacities, can be tuned under selected physical treatment conditions. Ultrasonication induces more prominent modifications to the protein structure, including partial unfolding, secondary structural changes, decreased crystallinity, and thermal alterations. While keeping the nutritional value of proteins intact, US and PEF treatments provide a promising non-thermal route for upgrading underutilized fava beans into value-added functional plant protein ingredients suitable to various food applications. Although this study provides useful insights, further work on the optimization of electric field strength, treatment time, and temperature needs finer tuning to maximize the derived functional improvements. Scale-up studies are also required to assess the feasibility and costs of such treatments at an industrial scale.

## Figures and Tables

**Figure 1 foods-13-00376-f001:**
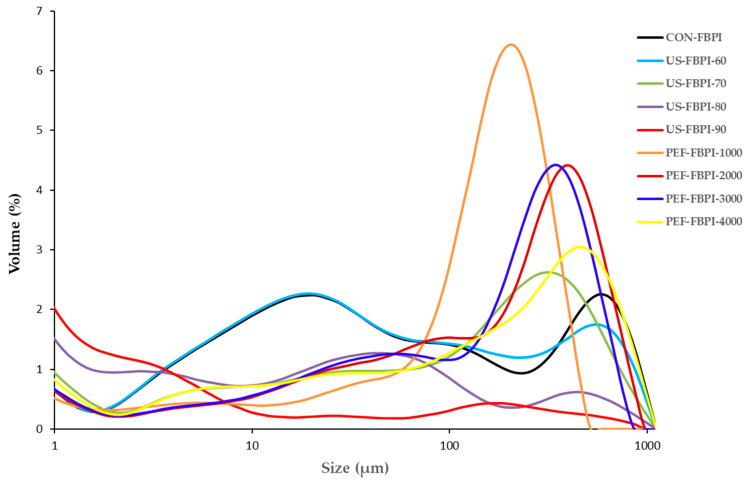
Particle size distribution of FBPIs treated with US and PEF at varying intensities. FBPI: fava bean protein isolate; US: ultrasonication; PEF: pulsed electric fields; CON-FBPI: native FBPI without any treatment; US-FBPI-60: FBPI ultrasonicated at 60% amplitude; US-FBPI-70: FBPI ultrasonicated at 70% amplitude; US-FBPI-80: FBPI ultrasonicated at 80% amplitude; US-FBPI-90: FBPI ultrasonicated at 90% amplitude; PEF-FBPI-1000: FBPI subjected to PEF treatment for 1000 pulses; PEF-FBPI-2000: FBPI subjected to PEF treatment for 2000 pulses; PEF-FBPI-3000: FBPI subjected to PEF treatment for 3000 pulses; and PEF-FBPI-4000: FBPI subjected to PEF treatment for 4000 pulses.

**Figure 2 foods-13-00376-f002:**
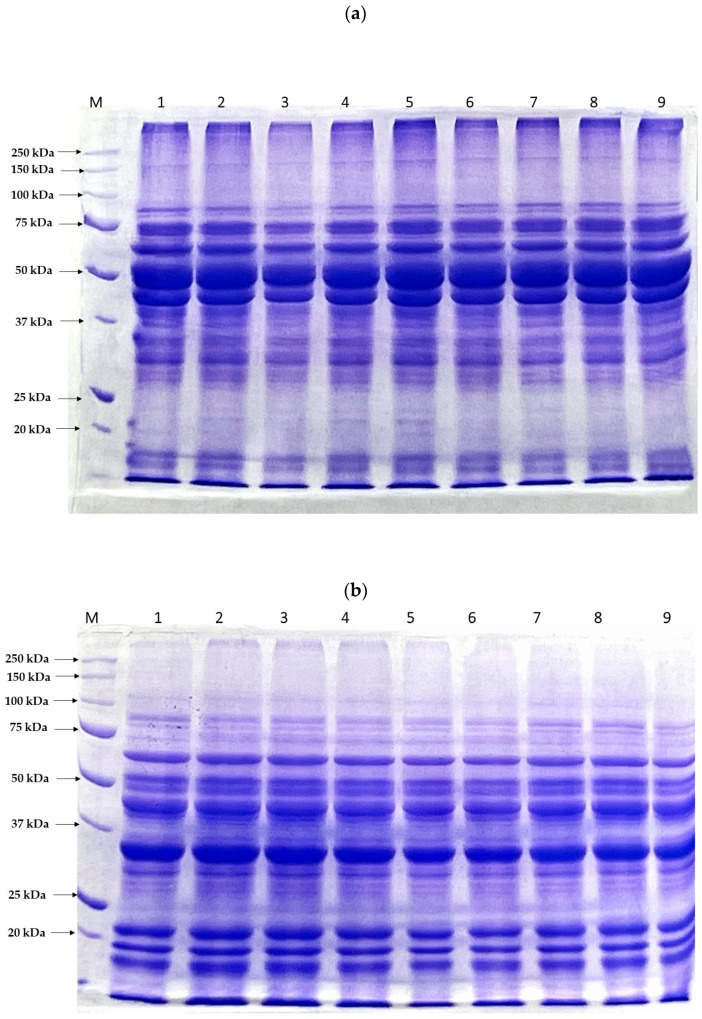
SDS-PAGE protein patterns of FBPIs treated with US and PEF at varying intensities in (**a**) the absence and (**b**) presence of β-mercaptoethanol. Lane M: molecular weight marker; Lane 1: CON-FBPI; Lane 2: US-FBPI-60; Lane 3: US-FBPI-70; Lane 4: US-FBPI-80; Lane 5: US-FBPI-90; Lane 6: PEF-FBPI-1000; Lane 7: PEF-FBPI-2000; Lane 8: PEF-FBPI-3000; and Lane 9: PEF-FBPI-4000. For the caption, please see Figure 1.

**Figure 3 foods-13-00376-f003:**
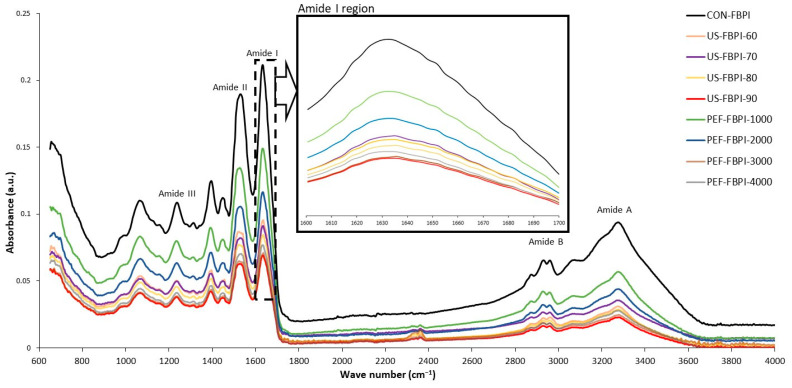
Fourier-transform infrared (FTIR) spectra of FBPIs treated with US and PEF at varying intensities. For the caption, please see Figure 1.

**Figure 4 foods-13-00376-f004:**
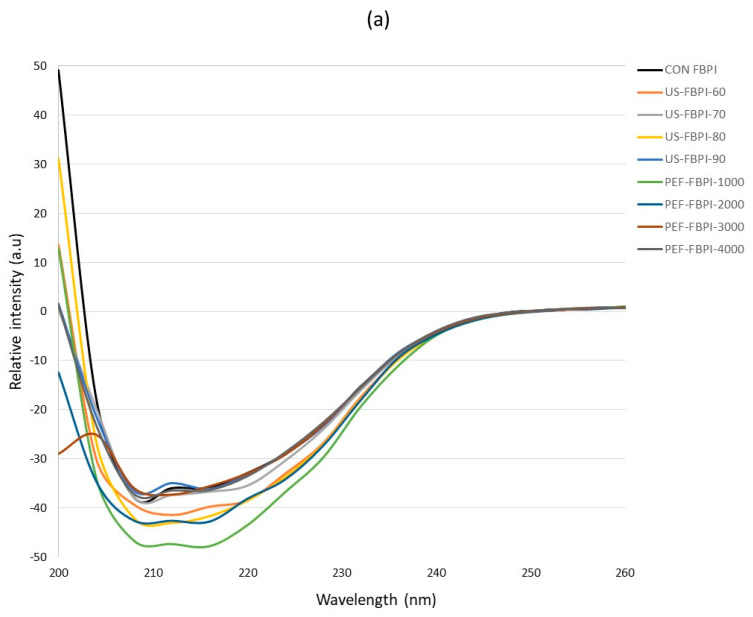
(**a**) Circular dichroism (CD) spectra and (**b**) relative content of secondary structure components in FBPIs treated with US and PEF at varying intensities. For the caption, please see Figure 1.

**Figure 5 foods-13-00376-f005:**
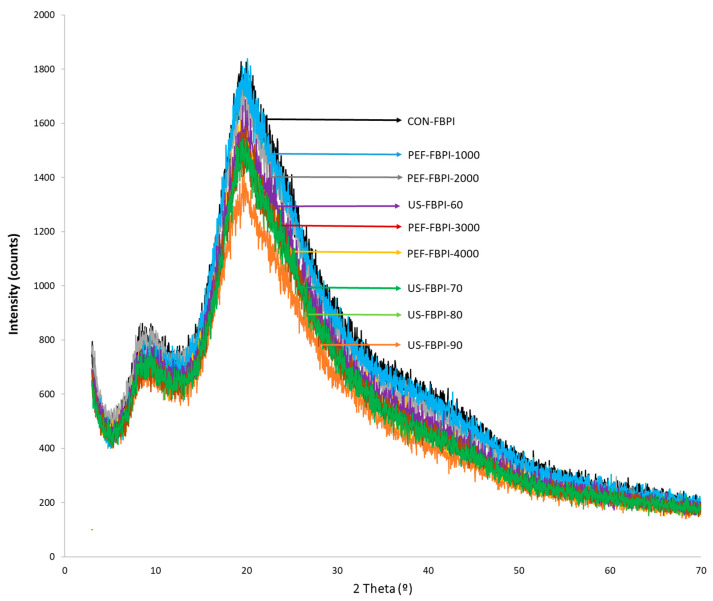
X-ray diffraction (XRD) spectra of FBPIs treated with US and PEF at varying intensities. For the caption, please see Figure 1.

**Figure 6 foods-13-00376-f006:**
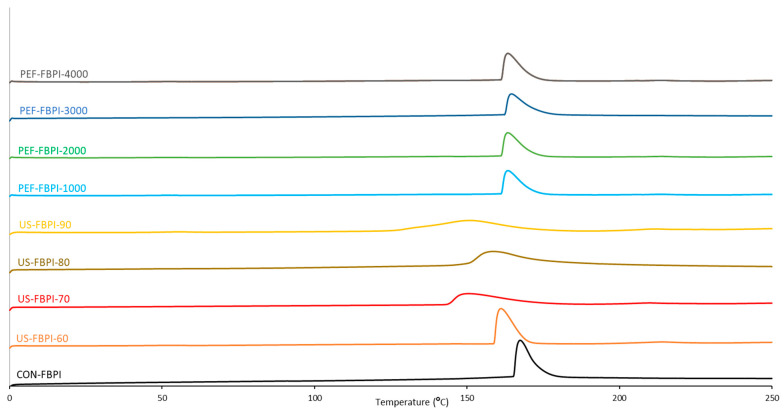
Differential scanning calorimetry (DSC) thermograms of FBPIs treated with US and PEF at varying intensities. For the caption, please see Figure 1.

**Table 1 foods-13-00376-t001:** Solubility, surface hydrophobicity, free sulfhydryl content, emulsifying activity index (EAI), and emulsion stability index (ESI) of FBPIs subjected to US and PEF treatments and varying intensities.

Sample	Solubility (%)	Surface Hydrophobicity	Free Sulfhydryl Content (µmol/g Protein)	EAI (m^2^/g Protein)	ESI (min)	Foaming Ability (%)	Foaming Stability (%)
CON-FBPI	74.38 ± 2.16 c	5625 ± 191 e	10.36 ± 0.34 f	34.90 ± 3.18 d	39.47 ± 5.26 c	74.44 ± 9.26 c	52.19 ± 3.62 e
US-FBPI-60	87.63 ± 2.89 a	7923 ± 204 b	11.76 ± 0.41 e	47.75 ± 4.42 c	43.30 ± 3.98 bc	119.97 ± 12.59 bc	78.83 ± 6.18 bc
US-FBPI-70	90.98 ± 2.67 a	9459 ± 336 a	17.08 ± 0.28 a	69.14 ± 4.34 a	55.13 ± 4.27 a	198.96 ± 11.75 a	102.74 ± 7.21 a
US-FBPI-80	80.53 ± 2.36 b	7422 ± 223 c	15.91 ± 0.27 b	58.65 ± 5.16 b	47.88 ± 2.97 b	135.33 ± 10.87 b	83.33 ± 8.92 b
US-FBPI-90	78.91 ± 2.81 bc	7589 ± 195 bc	15.71 ± 0.31 b	54.23 ± 3.89 bc	42.23 ± 5.21 bc	132.63 ± 8.21 b	82.61 ± 6.31 b
PEF-FBPI-1000	81.82 ± 2.98 b	7386 ± 208 c	13.51 ± 0.22 c	52.78 ± 6.67 bc	42.92 ± 2.25 bc	103.36 ± 15.78	63.34 ± 4.12 d
PEF-FBPI-2000	82.12 ± 1.99 b	7894 ± 301 b	13.56 ± 0.28 c	54.65 ± 3.27 bc	45.55 ± 3.64 bc	138.89 ± 13.34 b	72.72 ± 5.33 bcd
PEF-FBPI-3000	79.15 ± 2.25 b	6691 ± 197 d	11.34 ± 0.33 e	54.30 ± 6.91 bc	42.98 ± 4.18 bc	121.35 ± 6.63 bc	79.63 ± 7.43 bc
PEF-FBPI-4000	78.86 ± 2.37 bc	6568 ± 257 d	12.75 ± 0.19 d	53.95 ± 6.53 bc	43.66 ± 4.21 bc	122.58 ± 9.98 bc	68.16 ± 5.28 cd

Note: FBPI: fava bean protein isolate; US: ultrasonication; PEF: pulsed electric fields; CON-FBPI: native FBPI without any treatment; US-FBPI-60: FBPI ultrasonicated at 60% amplitude; US-FBPI-70: FBPI ultrasonicated at 70% amplitude; US-FBPI-80: FBPI ultrasonicated at 80% amplitude; US-FBPI-90: FBPI ultrasonicated at 90% amplitude; PEF-FBPI-1000: FBPI subjected to PEF treatment for 1000 pulses; PEF-FBPI-2000: FBPI subjected to PEF treatment for 2000 pulses; PEF-FBPI-3000: FBPI subjected to PEF treatment for 3000 pulses; and PEF-FBPI-4000: FBPI subjected to PEF treatment for 4000 pulses. Data are presented as the mean ± SD (n = 3). Different lowercase letters in the same row indicate significant differences (*p* < 0.05).

**Table 2 foods-13-00376-t002:** Particle size parameters and zeta potential of FBPIs subjected to US and PEF treatments and varying intensities.

Sample	Particle Size Parameters (µm)		Zeta Potential
			(mV)
	D[4,3]	D[3,2]	
CON-FBPI	72.6 ± 5.1 d	1.20 ± 0.02 a	−26.27 ± 0.54 a
US-FBPI-60	70.9 ± 6.1 d	1.13 ± 0.06 a	−24.27 ± 0.39 b
US-FBPI-70	61.2 ± 4.9 d	0.69 ± 0.04 c	−22.47 ± 0.17 c
US-FBPI-80	44.7 ± 2.8 e	0.5 ± 0.02 d	−22.40 ± 0.33 c
US-FBPI-90	25.65 ± 2.7 f	0.39 ± 0.01 e	−20.10 ± 0.86 d
PEF-FBPI-1000	144.1 ± 7.9 b	1.17 ± 0.16 a	−25.73 ± 1.27 a
PEF-FBPI-2000	162.3 ± 8.7 a	0.92 ± 0.04 b	−25.07 ± 0.66 ab
PEF-FBPI-3000	164.3 ± 8.9 a	0.83 ± 0.02 b	−25.87 ± 0.53 a
PEF-FBPI-4000	120.6 ± 7.2 c	0.81 ± 0.02 b	−25.57 ± 0.78 a

Note: FBPI: fava bean protein isolate; US: ultrasonication; PEF: pulsed electric fields; CON-FBPI: native FBPI without any treatment; US-FBPI-60: FBPI ultrasonicated at 60% amplitude; US-FBPI-70: FBPI ultrasonicated at 70% amplitude; US-FBPI-80: FBPI ultrasonicated at 80% amplitude; US-FBPI-90: FBPI ultrasonicated at 90% amplitude; PEF-FBPI-1000: FBPI subjected to PEF treatment for 1000 pulses; PEF-FBPI-2000: FBPI subjected to PEF treatment for 2000 pulses; PEF-FBPI-3000: FBPI subjected to PEF treatment for 3000 pulses; and PEF-FBPI-4000: FBPI subjected to PEF treatment for 4000 pulses. Data are presented as the mean ± SD (n = 3). Different lowercase letters in the same row indicate significant differences (*p* < 0.05).

**Table 3 foods-13-00376-t003:** Thermal transition temperatures and phase change enthalpy values of FBPIs subjected to US and PEF treatments and varying intensities.

Sample	Onset (°C)	Peak (°C)	Endset (°C)	Enthalpy (J/g)
CON-FBPI	164.25	166.63	179.17	191.86
US-FBPI-60	158.67	160.15	168.97	179.87
US-FBPI-70	143.85	150.09	173.17	119.40
US-FBPI-80	148.66	159.41	177.66	120.66
US-FBPI-90	128.89	150.48	172.09	112.75
PEF-FBPI-1000	160.31	162.85	173.01	147.71
PEF-FBPI-2000	161.89	164.84	174.58	152.50
PEF-FBPI-3000	163.20	165.10	178.92	144.97
PEF-FBPI-4000	161.16	162.57	171.45	145.70

Note: FBPI: fava bean protein isolate; US: ultrasonication; PEF: pulsed electric fields; CON-FBPI: native FBPI without any treatment; US-FBPI-60: FBPI ultrasonicated at 60% amplitude; US-FBPI-70: FBPI ultrasonicated at 70% amplitude; US-FBPI-80: FBPI ultrasonicated at 80% amplitude; US-FBPI-90: FBPI ultrasonicated at 90% amplitude; PEF-FBPI-1000: FBPI subjected to PEF treatment for 1000 pulses; PEF-FBPI-2000: FBPI subjected to PEF treatment for 2000 pulses; PEF-FBPI-3000: FBPI subjected to PEF treatment for 3000 pulses; and PEF-FBPI-4000: FBPI subjected to PEF treatment for 4000 pulses. Data are presented as the mean ± SD (n = 3).

## Data Availability

Data is contained within the article.

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
