# Peer review of "Tailoring the Techno-Functional Properties of Fava Bean Protein Isolates: A Comparative Evaluation of Ultrasonication and Pulsed Electric Field Treatments"

_foods, 2024, doi:10.3390/foods13030376_

Round 1

Reviewer 1 Report

Comments and Suggestions for Authors

The present study investigated the effects of ultrasonication and pulsed electric field treatments on the techno-functional properties of fava bean protein isolates. Overall, I think this manuscript is interesting, rich in data, and well-written. Therefore, I suggest to make some modifications as follows:

1. Lines 6-7: The format is a little informal. The information of province/state and country is missing.

2. Materials: The information of chemicals used in this study, including key reagents and enzmes, should also be provided with their suppliers and purity levels.

3. Table 2: Mean particle size? Sometimes the mean/average particle size could not present the actual size profile of samples. Strongly suggest to provide a figure illustrating the particle size distribution of tested samples.

4. Fig. 2: Commonly, the absorbance should be normalized to 0-1. Additionally, other chemical bonds excluding amide bands should also be marked in this figure.

5. References: Some cited literatures are too old, even published within the last century (before year 2000). Strongly suggest to update these old literaturtes.

6. English language is fine and only minor editing is required.

Comments on the Quality of English Language

English-writing is fine and only minor editing is required.

Author Response

Thank you for your constructive comments. Please see the attachment.

Reviewer 2 Report

Comments and Suggestions for Authors

The article Tailoring the techno-functional properties of fava bean protein 2 isolate: A comparative evaluation of ultrasonication and pulsed electric field treatments is well-written but lacking some important points, it is unacceptable in its present form and need to be revised

Comment: why these both technologies were applied together, as earlier studies there are number of studies have been published?

Comment: Need to improve the novelty of this with better logical limitation in earlier studies?

Comment: What could be the practical application of study?

Comment: Moreover, in my opinion the practical application of this study is very limited due to it advanced technologies and could be higher cost

Comment: There are some minor English mistakes need to be addressed

Comment: Overall, the research design is good

Comment: Section 3.1.3. need to add the more logical discussion in this part

Comment: Add the future recommendations and limitations of this study in the conclusion

Comments on the Quality of English Language

Moderate improvement is required

Author Response

Thank your for your comments. Please see the attachment.

Reviewer: 2
Comments and Suggestions for Authors

The article Tailoring the techno-functional properties of fava bean protein 2 isolate: A comparative evaluation of ultrasonication and pulsed electric field treatments is well-written but lacking some important points, it is unacceptable in its present form and need to be revised

***** Authors would like to thank the reviewer for taking the precious time to review the paper. All queries, including the ones pinned in the pdf, have been responded and the corrections have been made as highlighted in blue

Comment: why these both technologies were applied together, as earlier studies there are number of studies have been published?

*****So far there is no report on the use of Pulsed electric field for the modification of fava bean proteins. Although the studies on the use of ultrasound modification of FBPI are available, nevertheless, we wanted to see the comparative analysis of both the non-thermal technologies and figure out which one is more feasible and potent.

Comment: Need to improve the novelty of this with better logical limitation in earlier studies?

*****The authors have presented a novel study on the use of PEF for FBPI modification, which has never been done before. However, the novelty of this study has been clearly defined and revised. Please see lines 75-82. Thank you!

Comment: What could be the practical application of study?

*****Practical application of this study involves the use of these non-thermal techniques to induce more favorable properties such as gelling, emulsifying and foaming agents. On a larger scale the authors are working on the application of FBPI as plant-based meat analogues, where better solubility and gelling ability are desirable.

Comment: Moreover, in my opinion the practical application of this study is very limited due to it advanced technologies and could be higher cost

*****The application of PEF and ultrasonication technology is increasingly being commercialized for non-thermal food processing applications. However, scale-up and cost-optimization efforts are still needed to realize the full industrial potential of these technologies. A techno-economic analysis comparing the capital and operating costs of pilot and production-scale PEF and ultrasonication systems could provide useful insights for process design and investment planning. Understanding the energy inputs, throughputs achievable, and overall cost-competitiveness of PEF and ultrasound for protein modification tasks like those demonstrated in this study for FBPI would support technological maturation and commercialization pathways. Such an analysis, grounded in the fundamental understanding of processing-structure-property relationships gained through bench-top research, could aid in process translation efforts.

Comment: There are some minor English mistakes need to be addressed

*****The manuscript has been revised thoroughly for English language errors using Grammarly software.

Comment: Overall, the research design is good

*****Thank you!

Comment: Section 3.1.3. need to add the more logical discussion in this part

*****More discussion in this section has been added. Please see lines 363-368

Comment: Add the future recommendations and limitations of this study in the conclusion

*****Some recommendations and limitations have been added. Please see lines 644-647.

Reviewer 3 Report

Comments and Suggestions for Authors

The study reports on the impact of US and PEF on the functionality of faba bean protein isolate. While there would be more value in looking into a commercial isolate the approach is still sound as long as the authors remain focused on the science behind. 

There are however several issues, the first of them is use of an amplitude for US, for someone to repeat this the authors must use the energy density instead. This depends on the power drawn, time and the volume of the sample. So unless someone has everything that was available to these authors there is almost no chance this study could be repeated by someone outside of that laboratory.

Solubility is calculated wrong as it is always expressed relative to the initial protein concentration in the bulk. 

The additional queries can be found in the attached document.

Comments on the Quality of English Language

The submission is a bit too wordy and could be shortened but overall not too bad.

Author Response

Thank you very much for the timely comments. Please see the attachment with responses.

Round 2

Reviewer 2 Report

Comments and Suggestions for Authors

The comments are well-addressed and article can be considered for publication

Comments on the Quality of English Language

Minor changes are required

Reviewer 3 Report

Comments and Suggestions for Authors

The author's have addressed my queries

Comments on the Quality of English Language

Some editing required which can be done during the type setting.